# Global Trends of Nutrition in Cancer Research: A Bibliometric and Visualized Analysis Study over the Past 10 Years

**DOI:** 10.3390/ijerph19074165

**Published:** 2022-03-31

**Authors:** Bo-Young Youn, Seo-Yeon Lee, Wonje Cho, Kwang-Rok Bae, Seong-Gyu Ko, Chunhoo Cheon

**Affiliations:** 1Department of Preventive Medicine, Kyung Hee University, Seoul 02447, Korea; james_youn@khu.ac.kr (B.-Y.Y.); epiko@khu.ac.kr (S.-G.K.); 2Department of Science in Korean Medicine, Graduate School, Kyung Hee University, Seoul 02447, Korea; sylee225@khu.ac.kr; 3Department of Korean Medicine, Graduate School, Kyung Hee University, Seoul 02447, Korea; wj8090@hotmail.com; 4Department of Clinical Korean Medicine, Graduate School, Kyung Hee University, Seoul 02447, Korea; chall21cc@naver.com

**Keywords:** nutrition, nutrition therapy, cancer care, cancer management, bibliometric

## Abstract

The increasing application of nutrition in cancer management has attracted a great deal of research interest in recent decades. Nutritional therapies, interventions, and assessments were known to have positive effects on reducing side effects from cancer therapy. In order to identify the global research output for nutrition in cancer research, a bibliometric analysis during the past 10 years was conducted to evaluate the current status of trends, gaps, and research directions as no bibliometric studies have been conducted regarding nutrition and cancer. After the data collection, a total of 1521 articles were chosen for this bibliometric study. The visualization analysis was performed with VOSviewer. The number of publications has grown continuously since a substantial spark was identified in 2019. The majority of the authors’ affiliations were in European countries. Four cancer types were recognized among the top 10 author keywords; they were breast cancer, head and neck cancer, colorectal cancer, and gastric cancer. The Nutrients journal was the most popular among the authors as the journal published 195 articles related to the topic. In conclusion, providing evidence-based nutritional solutions for various types of cancer is essential to nutrition and cancer research. Since it is presumed to have a growing number of cancer patients worldwide with the aging population, it is vital to continuously generate research finding effective nutrition therapies for cancer patients.

## 1. Introduction

Nutrition therapy is a nutritional-based treatment, including identifying a person’s nutritional status and giving the right foods or nutrients to treat various diseases such as cancer, diabetes, and metabolic syndrome. Numerous studies reported that deficiency of certain nutrients proceeds brain dysfunction; moreover, poor nutritional status is associated with the prevalence of pre-metabolic syndrome, hospital stay, and the life expectancy of diabetes patients [1,2,3,4]. Based on the research, the supply of appropriate nutrients is vitally essential to curing and managing various diseases.

Cancer is a complex disease characterized by the uncontrolled growth of abnormal cells. The effect of malnutrition on health and cost factors in cancer patients has been shown in several studies; especially, cancer patients with malnutrition conditions showed poor outcomes, higher medical costs, and low quality of life [5,6,7]. With the increasing evidence, nutrition therapies, interventions, and assessments have shown its positive effects for managing various cancer types. For instance, studies have shown that the use of nutrition therapy results in better outcomes for patients with head and neck cancer, colorectal cancer, and pancreatic cancer [8,9,10].

Malnutrition is frequent in cancer patients, and the risk is higher in older patients or those treated with chemotherapy [11,12]. Aging is one of the most important risk factors for cancer. The rapid growth of the aging population is expected as the number of people with cancer aged 80 or older is expected to triple by 2050 [13]. More importantly, in older patients with cancer, malnourished patients had higher mortality rates than those who were well-nourished. Additionally, the elderly patients are particularly vulnerable to the impact of malnutrition caused by age-related changes and cancer-induced factors [14]. Thus, an early assessment of malnourishment is critical for selecting personalized nutritional interventions while preventing sudden nutritional deterioration [15].

Chemotherapy and radiotherapy are known as the elementary tumor therapeutic strategies used in most cancer cases. However, chemotherapy and radiotherapy often cause various side effects, which are highly susceptible to a malnourished status. Commonly reported side effects from cancer therapies include a decreased appetite, weight loss, fatigue, hair loss, nausea, and vomiting [16,17]. It is noteworthy that the recent research has outlined a positive role of nutritional therapy against side effects of cancer patients receiving chemo- and radiotherapy [18,19].

In the past, there was a lack of awareness of the clinical benefits and authoritative guidelines of nutritional therapy in cancer treatment due to insufficient evidence-based research on this topic. However, there have been outstanding efforts to promote nutrition for cancer patients. A guideline for nutrition in cancer patients was introduced on a continental level; the evidence-based guidelines were developed by ESPEN by the European Partnership for Action Against Cancer as an EU-level initiative [20]. A comprehensive review indicated that the nutritional guidelines had shown positive results of a decrease in various types of cancer risks [21].

Despite the significant interest in utilizing nutrition methodologies for cancer, there has not been enough scientific evidence to give more options for healthcare professionals to consider nutritional interventions [22]. Therefore, a bibliometric study of the global trend of nutrition-cancer research was considered.

As no bibliometric studies have been conducted to seek the global trends regarding nutrition and cancer, a bibliometric study was carried out over the past 10 years. This bibliometric analysis is essential to evaluate trends, gaps, and research directions in the area of interest; in addition, it is exceptionally effective in grasping the overall trend of the research activity [23]. Hence, the aim of the study was to perform a bibliometric analysis to explore the current status of research on nutrition and cancer from 2011 to 2021 to global research productivity, promoting future research priorities. In particular, the analysis followed the dimensions: (i) number of publication/years; (ii) authors; (iii) affiliations; (iv) author keywords; (v) journals; (vi) most cited publications; (vii) collaboration among countries.

## 2. Materials and Methods

A bibliometric analysis, a valuable tool for navigation in a particular research area, was conducted to identify key trends and patterns in the field of scientific data by applying mathematical and statistical methods to analyze the course of publications [24,25].

### 2.1. Data Source and Search Strategy

For bibliometric and visualization analysis, the metadata from the SCOPUS database were retrieved on 17 January 2022, to conduct analysis. The SCOPUS database was chosen because it is one of the largest databases for finding quality global journals [26]. Additionally, the SCOPUS database covers a broader range of topics than other indexes; it is more accessible to the public, allowing free access to author and source information, including metrics [27].

The search terms, related to (1) nutrition and (2) cancer, were chosen based on preliminary search, looking at various systematic reviews for relevant keywords [28,29,30,31], discussions among authors and suggestions from physicians and nutritionists. A detailed search query was outlined in Box 1.

Box 1The Search Query.(1)“nutrition” OR “nutritional therap*” OR “nutritional assessment” OR “nutritional supplement*” OR “nutritional intervention*”(2)“cancer” OR “cancer care” OR “tumor” OR “neoplasm*” OR
“malignanc*” OR “oncology” OR “cancer patient*” OR “cancer management” OR “cancer prevention”

### 2.2. Data Collection

The study period was from 2011 to 2021. To raise the accuracy of finding the relevant studies, the search query was performed in the “title” section only. Based on the search strategy, a total of 2040 articles were identified during the initial search. Then, the articles were restricted to those that (1) were written in English, (2) were original and review articles regardless of the type of animal or human studies, and (3) were in the final publication and in press. Any non-peer reviewed articles were excluded as the results may not be evidence-based and fallacious [32]. A title-specific search was carried out to identify highly cited articles [33]. Additionally, only English language publications were considered, as previous bibliometric studies indicated that the original articles written in English were the majority of the assessed literature [34]. Therefore, a final set of 1521 articles were selected in the final analyses (Figure 1).

### 2.3. Data Analysis and Visualization

The metadata was manually checked by two researchers (W.C., K.-R.B.) for any duplicates and errors, and then, imported to Rstudio v.4.0.2 software(R Studio, PBC, Boston, MA, USA) (22 June 2020) with the bibliometric R-package (http://www.bibliometrix.org) (accessed on 15 January 2022) was used to extract authors, author keywords, author collaboration in between countries, affiliated institutions, journals, citations, and countries; in addition, the word cloud and world map of collaborative authors were created with the Biblioshiny app from the bibliometric package [35]. Bibliometrix and biblioshiny are the open-source packages used in the R language environment. Bibliometrix is capable of the entire process of scientific data analysis, while biblioshiny enables users to create a visual analysis based on an interactive web interface [35]. The author keyword co-occurrences and cluster keywords were analyzed and visualized using the VOSviewer v.1.6.11 (Centre for Science and Technology Studies, Leiden University, Leiden, The Netherlands). The VOSviewer has been used in numerous previous bibliometric studies for creating co-occurrence matrix and identifying clusters from the keywords network [36,37]. The global trend of publications graph was created using Microsoft Excel 2020.

## 3. Results

### 3.1. The Global Publication Trend

The number of publications regarding nutrition and cancer has grown over a decade. The average growth rate was 10.03%, and the highest number of publications was in 2021 with a total of 268 articles. According to Figure 2, there was a substantial spark in publications since 2019 compared to any previous years.

### 3.2. Authors, Author Keywords, Affiliations

Tumino has generated the highest number of articles (195 articles; 12.82% of the total number of articles) among the 7134 authors; the second highest was Trichopoulous (191, 12.56%; Table 1). With regard to the author affiliations, the top 10 list shows that the authors belonged to institutions in European countries (Table 2). Among the 10 author affiliations, three were in United Kingdom (Imperial College London—London, University of Oxford—Oxford, and University of Cambridge—Cambridge, England). The authors from the Imperial College London have published the highest number of articles so far (410 articles).

Of the 2398 author keywords, the top 10 keywords were identified in Table 3. The key word with the highest number of occurrences was nutrition (236 occurrences); the second highest was cancer (202 occurrences). Amid the list, four different types of cancer were found: breast cancer, head and neck cancer, colorectal cancer, and gastric cancer. Breast cancer had the highest occurrence among the cancer types with the 71 occurrences.

As Figure 3 illustrates, the author keywords were divided into four clusters with four different colors: red (cluster 1), green (cluster 2), blue (cluster 3), and yellow (cluster 4). The clusters are the indication of the major research directions; the size of the nodes indicates the frequency of occurrence. The curves between the nodes represent the co-occurrences in the same publication. The shorter the distance between the two nodes, the larger the number of co-occurrences of the two keywords.

In order to depict the frequency of author keywords, a word-cloud was created to find more relevant keywords besides the top 10 keywords (Figure 4). The font size represents the frequency of occurrence, and the important keywords apart from the top 10 list were found, such as lung cancer, sarcopenia, esophageal cancer, prostate, pancreatic cancer, etc. Based on the word-cloud, it can be presumed that the utilization of nutrition could be adapted to a wide array of types of cancer.

### 3.3. Journals

Of the 516 journals, the Nutrients journal of the Multidisciplinary Digital Publishing Institute was found to have the highest number of articles (70 articles); the second highest journal was the International Journal of Cancer (68 articles). Each journal had a different publisher, but the proposed aim and scope were similar, heavily focusing on generating evidence-based findings in nutrition-cancer research. The details are described in Table 4.

### 3.4. Country Collaboration

Although the majority of the author affiliations was in European countries, the corresponding authors from the United States have contributed the most articles (239 articles). The corresponding authors from China were ranked in second (170 articles). Despite the ranking, five European countries were ranked in the top 10 list (Table 5).

A world collaboration map was generated to discover the world’s research. The blue color on the map represents the number of publications per country with intensity; the red lines indicate the extent of collaboration between the authors. It is notable that authors from Italy have vibrantly collaborated with authors from Germany, United Kingdom, France, and Spain (Table 6, Figure 5). The second most active authors were from United Kingdom; the authors collaborated with authors from Germany, France, and Spain.

## 4. Discussion

This bibliometric analysis on the topic of nutrition and cancer has provided profound findings to consider. First of all, the number of publications has been upward with the substantial growth identified since 2019. As more than 100 publications were increased since 2019, it can be expected to have more than 300 publications in 2022. The trend will continue to rise as cancer is more susceptible to aging population and is one of the leading causes of death in males and females aged 60 to 79 years [38].

Surprisingly, a majority of the top authors’ affiliations were located in Europe. This perhaps could be associated with the high prevalence of use of complementary and alternative medicine (CAM). Kemppainen et al. mentioned that 25.9% of the general population in Europe has used CAM modalities in 2017 [39]. The results can be seen oddly since there is a higher prevalence of using traditional medicine in Asian countries since the traditional medicine is embedded in health systems in Korea, Japan, and China [40]. This may be perhaps cancer patients are more inclined to use herbal medicines rather than nutrition [41].

With regard to the author keywords, malnutrition was the highest after the nutrition and cancer keywords. Even though nutritional status is regularly examined throughout the patients’ treatment, an accurate nutritional screening is critical. An acute nutritional evaluation ought to be performed at the time of each cancer diagnosis [42]. In addition, malnutrition and eating disorders are common features during and beyond cancer therapies so that continuous nutritional management can be the key for managing cancer [43].

From the analysis of the author keywords, breast cancer, head and neck cancer, colorectal cancer, and gastric cancer were within the top 10 list; however, it can be assumed that nutrition is well-applied to various types of cancer treatments from the results on the word cloud visualization. Other than the mentioned cancer types, lung cancer, pancreatic cancer, gastrointestinal cancer, and esophageal cancer were revealed. The reason is that nutrition influences key cellular and molecular processes that characterize cancer cells; therefore, good nutrition is especially important as it can change the illness and the course of treatments [44].

Regarding the analysis of country collaboration, it is highly one-sided since the promotion of nutrition for cancer is heavily practiced in European countries. Some countries from Asia and North America were identified; however, not enough collaboration occurred to provide in-depth research. A collaborative effort between Asian countries is highly recommended, as the Asian countries have been used herbal medicines for a long time. Furthermore, a collaboration with the African countries is much needed since the concept of nutrition therapies are not much promoted [45].

This study has a few limitations. First, using only the literature in English may have ignored vital insights from the literature in other languages. Considering only English publications, a few important articles in other languages may have been missed. Since a bibliometric analysis does not address specific research questions, narrow areas of the topic are not identified.

Despite the limitations, future research directions having broader exploratory research questions aimed at mapping keywords and finding gaps in the related existing knowledge and research are distinguishable using a bibliometric analysis. Hence, this bibliometric analysis contributed to a better understand of nutrition and cancer research in a clear way with the vast amount of literature with the key elements.

## 5. Conclusions

This bibliometric and visualized analysis of nutrition in cancer research revealed that the global trend of research regarding nutrition and cancer has gradually increased. As the number of publications has sparked since 2019, it is expected that the research output will increase in the near future. Based on the analysis of the word-cloud using the author keywords, research applying nutrition in various types of cancer could be effective. Research collaboration was noted worldwide; however, authors from European countries vibrantly worked together. Since the topic was set to be rather broad, further research should highlight in much narrower topics, indicating certain nutritional therapies and interventions with cancer patients. The findings of this study will be insightful for researchers and practitioners working in the field of nutrition and cancer, illuminating and suggesting future research directions for the betterment of cancer care, providing evidence-based solutions for healthcare professionals.

## Figures and Tables

**Figure 1 ijerph-19-04165-f001:**
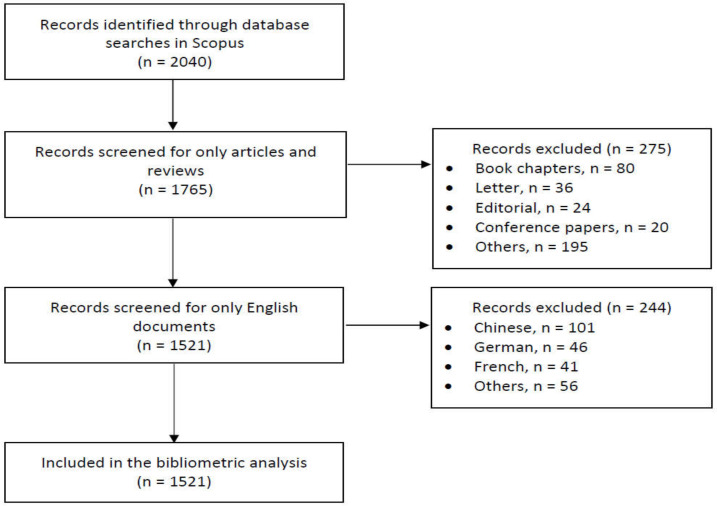
Flow diagram for the bibliometric analysis.

**Figure 2 ijerph-19-04165-f002:**
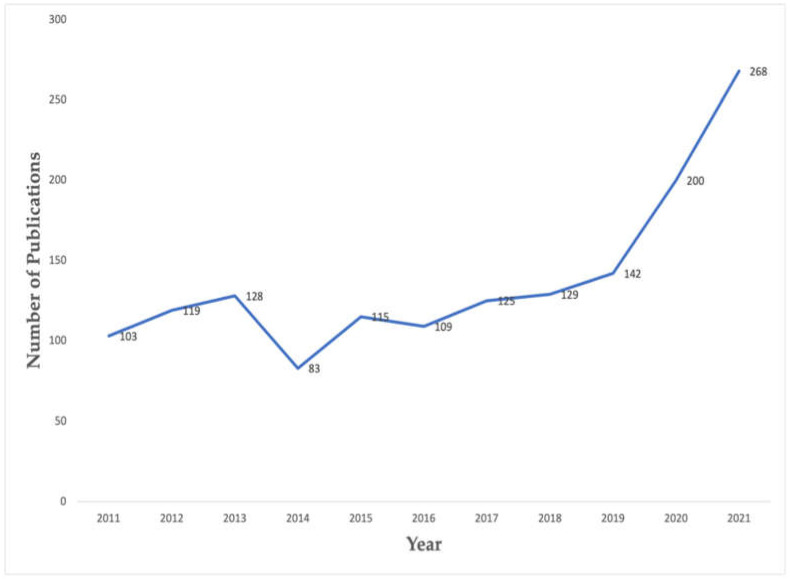
The number of publications on nutrition and cancer from 2011 to 2021.

**Figure 3 ijerph-19-04165-f003:**
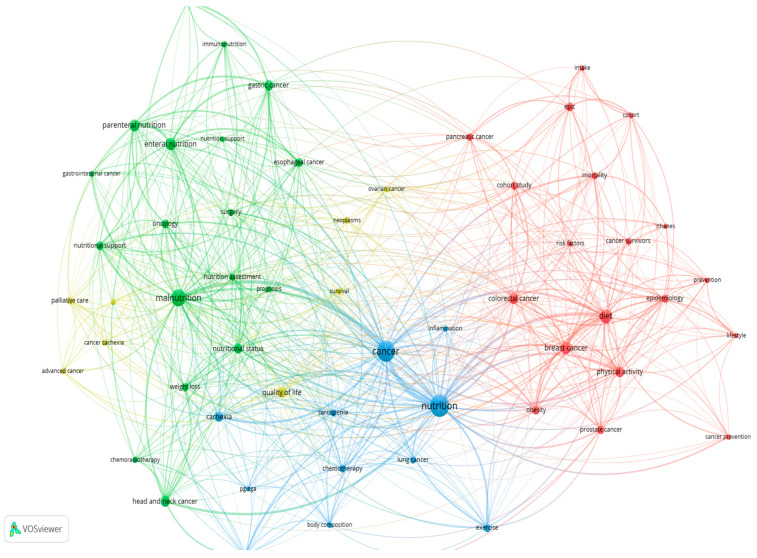
A network map of the co-occurrence analysis of the author keywords.

**Figure 4 ijerph-19-04165-f004:**
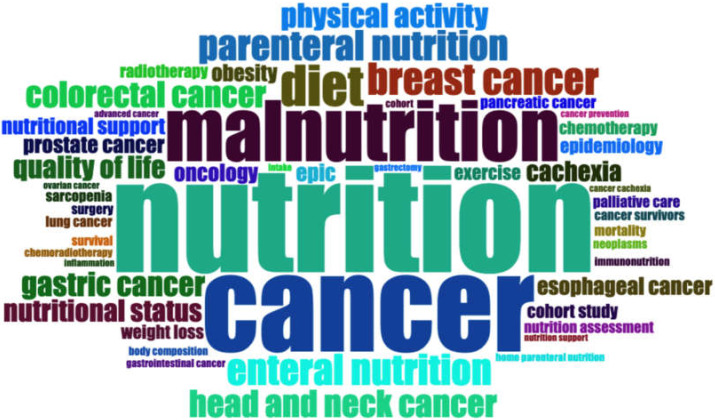
The word-cloud analysis on nutrition and cancer from 2011 to 2021.

**Figure 5 ijerph-19-04165-f005:**
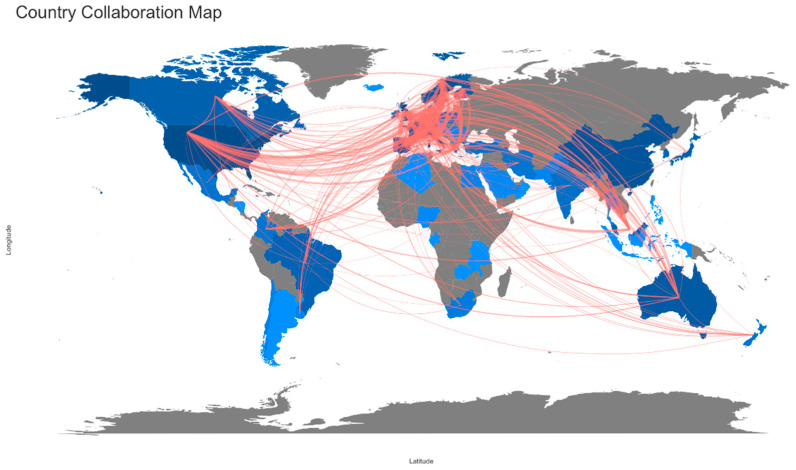
World map of country collaboration.

**Table 1 ijerph-19-04165-t001:** Top 10 most productive authors.

Rank	Author	No. of Articles	% of the Total Number of Articles (1521)
1	Tumino, R.	195	12.82%
2	Trichopoulou, A.	191	12.56%
3	Riboli, E.	179	11.77%
4	Overvad, K.	175	11.51%
5	Boeing, H.	170	11.18%
6	Tjønneland, A.	152	9.99%
7	Khaw, K-T.	141	9.27%
8	Kaaks, R.	135	8.88%
9	Bueno-De-Mesquita, HB.	129	8.48%
10	Palli, D.	117	7.69%

**Table 2 ijerph-19-04165-t002:** Top 10 most affiliation of the authors.

Rank	From	Country	Articles
1	Imperial College London	United Kingdom	410
2	International Agency for Research on Cancer	France	392
3	University of Oxford	United Kingdom	342
4	University of Athens Medical School	Germany	258
5	University of Cambridge	United Kingdom	250
6	University of Tromso	Norway	235
7	University Medical Center Utrecht	Netherlands	233
8	Umea University	Sweden	232
9	Aarhus University	Denmark	202
10	Lund University	Sweden	199

**Table 3 ijerph-19-04165-t003:** Top 10 list of the author keywords.

Rank	Author Keyword	Occurrences
1	Nutrition	236
2	Cancer	202
3	Malnutrition	135
4	Diet	91
5	Breast Cancer	71
6	Enteral Nutrition	69
7	Parenteral Nutrition	67
8	Head and Neck Cancer	60
9	Colorectal Cancer	58
10	Gastric Cancer	54

**Table 4 ijerph-19-04165-t004:** Top 10 most productive journals.

Rank	Journal	No. of Articles	Publisher	Aim and Scope
1	Nutrients	70	MDPI	Nutrients is an international, peer-reviewed open access advanced forum for publishing studies related to Human Nutrition.
2	International Journal of Cancer	68	Wiley	A broad scope of topics relevant to the following categories: Cancer Epidemiology, Cancer Genetics and Epigenetics, Infectious Causes of Cancer, Innovative Tools and Methods, Molecular Cancer Biology Tumor Immunology and Microenvironment, Tumor Markers and Signatures, and Cancer Therapy and Prevention.
3	Nutrition and Cancer	59	Taylor and Francis	This publication reports current findings on the effects of nutrition on the etiology, therapy, and prevention of cancer. Coverage of therapy focuses on research in clinical nutrition and oncology, dietetics, and bioengineering. Prevention approaches include public health recommendations, preventative medicine, behavior modification, education, functional foods, and agricultural and food production policies.
4	Supportive Care in Cancer	52	Springer	This journal covers the physical, psychosocial and spiritual aspects of the cancer patient’s journey from diagnosis through to end-of-life care, with a focus on interventions to mitigate symptoms and side effects during this journey.
5	American Journal of Clinical Nutrition	33	Oxford Academic	Well-controlled clinical studies that describe scientific mechanisms, efficacy, and safety of dietary interventions in the context of disease prevention or a health benefit will be considered. Public health and epidemiologic studies relevant to human nutrition, and innovative investigations of nutritional questions that employ epigenetic, genomic, proteomic, and metabolomic approaches are encouraged.
6	Clinical Nutrition	31	Elsevier	The journal reflects the scientific nature of this multidisciplinary background and encourages the coordination of investigation and research from these disciplines. It also publishes scientific works related to the development of new techniques and their application in the field of clinical nutrition.
7	British Journal of Nutrition	28	Cambridge Core	This journal focuses in nutritional science relevant to human or animal nutrition. The BJN welcomes studies in nutritional epidemiology, nutritional requirements, metabolic studies, body composition, energetics, appetite and obesity.
8	Journal of Parenteral and Enteral Nutrition	24	Aspen	This journal is the premier scientific journal of nutrition and metabolic support. It publishes studies that define the cutting edge of basic and clinical research in the field. It explores the science of optimizing the care of patients receiving enteral or IV therapies.
9	Cancer Epidemiology Biomarkers and Prevention	23	AACR	This journal publishes population-based research on cancer etiology, prevention, surveillance, and survivorship. The following topics are of special interest: descriptive, analytical, and molecular epidemiology; biomarkers including assay development, validation, and application; chemoprevention and other types of prevention research in the context of descriptive and observational studies; the role of behavioral factors in cancer etiology and prevention; survivorship studies; risk factors; implementation science and cancer care delivery; and the science of cancer health disparities.
10	BMC Cancer	20	BMC	BMC Cancer considers articles on all aspects of cancer research, including the pathophysiology, prevention, diagnosis and treatment of cancers. The journal also welcomes molecular and cellular biology, genetics, epidemiology, and clinical trials.

**Table 5 ijerph-19-04165-t005:** Top 10 corresponding authors’ countries.

Rank	Country	No. of Articles
1	United States of America	239
2	China	170
3	United Kingdom	120
4	Spain	97
5	Australia	85
6	Japan	82
7	Italy	77
8	France	68
9	Germany	62
10	Korea	49

**Table 6 ijerph-19-04165-t006:** Top 10 countries with the highest frequencies of collaboration.

Rank	From	To	Frequency
1	Italy	Germany	215
2	Italy	United Kingdom	215
3	France	Germany	214
4	United Kingdom	Germany	213
5	Italy	France	212
6	United Kingdom	France	211
7	Italy	Spain	210
8	Spain	France	209
9	Spain	Germany	207
10	United Kingdom	Spain	204

## Data Availability

The datasets used and/or analyzed during the current study are available from the first author upon reasonable request.

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
