# Peer review of "Global Trends of Nutrition in Cancer Research: A Bibliometric and Visualized Analysis Study over the Past 10 Years"

_ijerph, 2022, doi:10.3390/ijerph19074165_

Round 1

Reviewer 1 Report

Dear Authors

Lines 71 to 75 should be better in the end of your introduction to justify the necessity of your study.

Lines 92 to 98, data collection must be better clarified to explain clearly what is your logic to include and exclude the studies. Are animal studies included in the present study? Or was it limited to human studies? Review on the topic were included or only experimental studies? If you included review studies you must justify why (to find other experimental studies that are in your selective criterions or for other reason), culture cells studies were included or only studies with real patients with cancer? Our conviction is that you only selected studies with human cancer patients in treatment but it is not clearly explained.

We wish you success in the publication of your article

Author Response

Dear Reviewer:

Thank you very much for your time in reviewing our manuscript. Please review the comments below, and we also have highlighted (yellow) in our newest version of the manuscript. We have added more sentences along with proper references from the comments from other reviewers.

1) Lines 71 to 75 should be better in the end of your introduction to justify the necessity of your study.

-> Please review line 76-80, we have added a better description of bibliometric analysis.

2) Lines 92 to 98, data collection must be better clarified to explain clearly what is your logic to include and exclude the studies. Are animal studies included in the present study? Or was it limited to human studies? Review on the topic were included or only experimental studies? If you included review studies you must justify why (to find other experimental studies that are in your selective criterions or for other reason), culture cells studies were included or only studies with real patients with cancer? Our conviction is that you only selected studies with human cancer patients in treatment but it is not clearly explained.

-> Please review the line 111, we have added a sentence that regardless of animal or human studies.

We wish you success in the publication of your article

-> We really appreciate your time and help in making our manuscript enhance.

Respectfully,

Reviewer 2 Report

The manuscript suggests a current and attractive topic for academia. The effort made is evident, but it presents conceptual errors in the methodology, an inadequate search for terms and poor analysis, which do not allow it to meet the standards of bibliometric studies. I attach several observations that I hope will be useful in improving the manuscript:

Introduction

1) In general terms, the authors should revise your introduction; the literature presented in Nutrition in Cancer Research is very poor. It shows few references and a lot of unsupported text, which is incompatible for a prestigious journal. The few references used do not show authors of references in the study area. It should be noted that the introduction is very brief.

2) It should indicate the importance of the study presented for the academy. In other words, a bibliometric study.

3) It does not mention relevant studies in the area. For example:

- Arends, J., Bachmann, P., Baracos, V., Barthelemy, N., Bertz, H., Bozzetti, F., ... & Preiser, J. C. (2017). ESPEN guidelines on nutrition in cancer patients. Clinical nutrition, 36(1), 11-48.

- Ravasco, P. (2019). Nutrition in cancer patients. Journal of clinical medicine, 8(8), 1211.

- Donaldson, M. S. (2004). Nutrition and cancer: a review of the evidence for an anti-cancer diet. Nutrition journal, 3(1), 1-21.

Or literature reviews on the same topic:

- Mayne, S. T., Playdon, M. C., & Rock, C. L. (2016). Diet, nutrition, and cancer: past, present and future. Nature reviews clinical oncology, 13(8), 504-515.

4) There is no mention of the importance of bibliometric studies in this topic.

5) It is necessary to state how the article will be divided (number of sections) and what each of them comprises.

Materials and Methods

This section requires complete restructuring, expansion and new references (there are practically no references for a modern and fashionable topic). Bibliometric studies have a fundamental characteristic; they must be systematic, transparent and reproducible. As far as possible, a sequence of methodological stages (steps) and a figure summarising the process should be indicated.

6) Lines 71 - 74. It is necessary to explain with references why it is important to carry out a bibliometric study, that is to say, what this type of study brings me when the classic thing is to carry out a literature review. Use important references in the field.

7) Why use Scopus? There are other important databases such as Web of Science, Dimensions, and PubMed. Use arguments and references to defend your idea. That is, do not just expose one single idea.

8) There is a serious problem here. In lines 78 - 80, mention the use of Scopus, but in lines, 82 - 86 indicate the use of PubMed. There is an obvious contradiction. What are the systematic reviews used to choose the search keywords?

9) You should show the complete search equation so that the reader can repeat it.

10) Why do you use only titles (see line 93). It is usual to use "title, abstract and keywords" why did you do it differently. Explain and use references to studies with similar procedures.

11) The authors imply that they used articles. Why? Can you argue? Use references.

12) Use documents in English. Use references to validate what has been done.

13) You must indicate the usefulness of the software used. You must indicate: What are they? What is it for? Moreover, which relevant studies have used the software.

14) The methodology does not include data cleaning. Can you indicate this? Data cleaning is normally contemplated in bibliometric studies.

15) When applying inclusion and exclusion criteria, you should report the number of documents that remain in the process.

Results

16) Most analyses are purely descriptive, which makes the paper of no academic value. Presenting figures or tables and describing them is not research. Analyses should guide the reader to more detailed information about the manuscripts, authors, countries, and topics. Authors should provide an analysis of the content of each figure and indicate the significance of the data presented. I recommend exploring papers that do this type of bibliometric analysis; I attach an example of the hundreds of papers available online; these are MDPI and are recent according to the advances in bibliometrics (Giraldo et al. 2019 10.3390/agronomy9070352 or Carrion et al. 2021 10.3390/ijerph18189445). Basically, it should set out the contribution of each analysis to the academy. These articles set out a very appropriate structure for a Bibliometric analysis. They deal in a very interesting way with methodology, results and discussion. The authors present several studies in Bibliometrics that can serve as examples.

 Here are some examples:

17) Figure 1 shows errors. For example, the sum of publications is 1521, but line 94 shows 2040 articles. Moreover, in the table, it shows that ten authors wrote 1584. Evidently, some errors make the results questionable. 

18) Table 1 shows errors. If it shows ten authors whose total number of articles is greater than 1500 in the subject? Additionally, it would be interesting to know the author's university and other details that make the table and the analysis interesting.

19) The "co-occurrence" analysis of keywords (figure 2) is poor. The exposed label is wrong. Please review bibliometric studies. Here you should explain what the clusters and each node mean. Usually, the cluster determines the themes of the field of study and the nodes the topics. It would help if you elaborated on the presented studies to form the cluster. It is no use repeating the nodes you see.

20) Figure 3 is useless if you present table 3, as it is the same thing. The analyses about them are poor; their explanations are poor.

21) Table 4 is useless because you can see the same thing in figure 2.

22) Table 5 uses the Impact factor, a Web of Science indicator. It is assumed that the study used Scopus. Therefore the table is wrong. It should present more information regarding the journals.

23) It should indicate the usefulness of table 6. However, it does not state the contribution to the academy of knowing the main documents.

24) Table 7 is useless if you present figure 3. However, the information in table 7 can be used for analysis. For example, in figure 3, what does each colour and the lines represent?

25) In the analysis in figure 7, you should indicate. Whom does United States of America work with, are they related to the second and third place in any studies? What kind of themes do they share? Be creative and surprise your readers. In other words, explain with references the exposed studies of the existing relationships in table 8.

Discussion and conclusions

26) The discussion section lives up to its name because you discuss the results you have found here. Here, you should cross-check the figures and tables presented as generalisations of your study against the theory to determine whether there are aspects that can affirm existing theory or present new findings. Because of errors in the results, your discussion is dubious. It should be completely revised.

27) In relation to the conclusions, they should be brief and highlight important findings, limitations or future lines of research. As it stands, it is not relevant.

Author Response

Dear Reviewer:

We appreciate your time in reviewing our manuscript. Please review the newest version of our manuscript.

Introduction

1) In general terms, the authors should revise your introduction; the literature presented in Nutrition in Cancer Research is very poor. It shows few references and a lot of unsupported text, which is incompatible for a prestigious journal. The few references used do not show authors of references in the study area. It should be noted that the introduction is very brief.

-> We have added more references and sentences using your comment #3.

2) It should indicate the importance of the study presented for the academy. In other words, a bibliometric study.

-> Please review lines 76 to 80.

3) It does not mention relevant studies in the area. For example:

- Arends, J., Bachmann, P., Baracos, V., Barthelemy, N., Bertz, H., Bozzetti, F., ... & Preiser, J. C. (2017). ESPEN guidelines on nutrition in cancer patients. Clinical nutrition, 36(1), 11-48.

- Ravasco, P. (2019). Nutrition in cancer patients. Journal of clinical medicine, 8(8), 1211.

- Donaldson, M. S. (2004). Nutrition and cancer: a review of the evidence for an anti-cancer diet. Nutrition journal, 3(1), 1-21.

Or literature reviews on the same topic:

- Mayne, S. T., Playdon, M. C., & Rock, C. L. (2016). Diet, nutrition, and cancer: past, present and future. Nature reviews clinical oncology, 13(8), 504-515.

4) There is no mention of the importance of bibliometric studies in this topic.

-> Please review lines 76 to 80.

5) It is necessary to state how the article will be divided (number of sections) and what each of them comprises.

-> Please review lines 82 to 85.

Materials and Methods

6) Lines 71 - 74. It is necessary to explain with references why it is important to carry out a bibliometric study, that is to say, what this type of study brings me when the classic thing is to carry out a literature review. Use important references in the field.

-> Please review lines 88-90.

7) Why use Scopus? There are other important databases such as Web of Science, Dimensions, and PubMed. Use arguments and references to defend your idea. That is, do not just expose one single idea.

-> Please review lines 95 to 98.

8) There is a serious problem here. In lines 78 - 80, mention the use of Scopus, but in lines, 82 - 86 indicate the use of PubMed. There is an obvious contradiction. What are the systematic reviews used to choose the search keywords?

-> Please review lines 100.

9) You should show the complete search equation so that the reader can repeat it.

-> Please review Box 1. That is the complete search equation that we used along with the time frame.

10) Why do you use only titles (see line 93). It is usual to use "title, abstract and keywords" why did you do it differently. Explain and use references to studies with similar procedures.

-> Please review lines 113 to 114.

11) The authors imply that they used articles. Why? Can you argue? Use references.

-> Please review line 113.

12) Use documents in English. Use references to validate what has been done.

-> Please review lines 114 to 116, and also indicated in the limitations as well.

13) You must indicate the usefulness of the software used. You must indicate: What are they? What is it for? Moreover, which relevant studies have used the software.

-> Please review lines 130 to 133 / 136 to 137.

14) The methodology does not include data cleaning. Can you indicate this? Data cleaning is normally contemplated in bibliometric studies.

-> Please review lines 125 to 126.

15) When applying inclusion and exclusion criteria, you should report the number of documents that remain in the process.

-> We have created a flow diagram in Figure 1.

Results

 Here are some examples:

17) Figure 1 shows errors. For example, the sum of publications is 1521, but line 94 shows 2040 articles. Moreover, in the table, it shows that ten authors wrote 1584. Evidently, some errors make the results questionable. 

18) Table 1 shows errors. If it shows ten authors whose total number of articles is greater than 1500 in the subject? Additionally, it would be interesting to know the author's university and other details that make the table and the analysis interesting.

-> To address both 17 and 18, Please note that authors may have written papers together with other authors so that there are duplicates. Hence, the numbers do not add up the way you have described. We have tried to locate more details of authors; however, it was impossible to find each author's current position.

19) The "co-occurrence" analysis of keywords (figure 2) is poor. The exposed label is wrong. Please review bibliometric studies. Here you should explain what the clusters and each node mean. Usually, the cluster determines the themes of the field of study and the nodes the topics. It would help if you elaborated on the presented studies to form the cluster. It is no use repeating the nodes you see.

-> Please review lines 171 to 174.

20) Figure 3 is useless if you present table 3, as it is the same thing. The analyses about them are poor; their explanations are poor.

-> Please review lines 179-181 / 183-184.

21) Table 4 is useless because you can see the same thing in figure 2.

-> Deleted

22) Table 5 uses the Impact factor, a Web of Science indicator. It is assumed that the study used Scopus. Therefore the table is wrong. It should present more information regarding the journals.

-> Please review our new table and also review lines 191 to 193.

23) It should indicate the usefulness of table 6. However, it does not state the contribution to the academy of knowing the main documents.

-> deleted.

24) Table 7 is useless if you present figure 3. However, the information in table 7 can be used for analysis. For example, in figure 3, what does each colour and the lines represent?

-> Please review the lines 204 to 206.

25) In the analysis in figure 7, you should indicate. Whom does United States of America work with, are they related to the second and third place in any studies? What kind of themes do they share? Be creative and surprise your readers. In other words, explain with references the exposed studies of the existing relationships in table 8.

-> Please note that bibliometric analysis studies are quantitative studies, not qualitative studies. Therefore, it is impossible to locate every single exposed study. If it were to do so, it has to be either a systematic review or a scoping review study. With that said, we were not able to analyze further to locate studies.

Discussion and conclusions

26) The discussion section lives up to its name because you discuss the results you have found here. Here, you should cross-check the figures and tables presented as generalisations of your study against the theory to determine whether there are aspects that can affirm existing theory or present new findings. Because of errors in the results, your discussion is dubious. It should be completely revised.

-> We tried our best to reorganize and revise the discussion section.

27) In relation to the conclusions, they should be brief and highlight important findings, limitations or future lines of research. As it stands, it is not relevant.

-> We made it brief as you have advised.

Thank you for your time and help in enhancing our manuscript.

Respectfully,

Reviewer 3 Report

The review entitled: Global Trends of Nutrition in Cancer Research: A Bibliometric and Visualized Analysis Study Over the Past 10 Years presents scientific and practical importance.

The review is prepared according to the journal requirements and falls within its topic. The study is well documented, including a number of 37 bibliographic references. Terminology and nomenclature are correct and meet international standards.

Bibliometric analysis was used to explore the status and characteristics of publications in this field, including the most cited articles, research institutions, and authors. The 10-year trend in this field has been clarified, providing ideas and direction for researchers.

Here are some considerations:

- Section 2.2. Data Collection:

Should be put flowchart showing the progress of article selection.

- Section 3.2. Authors, Author Keywords, Affiliations:

Figures 2 and 3 are incorrectly titled.

- Section 3.3. Journals:

Table 6 has the wrong title.

I consider that the paper can be published after these changes.

Author Response

Dear Reviewer:

We appreciate your time in reviewing our manuscript. Please review the newest version of our manuscript. We also have added more sentences and references from the comments we received from other reviewers.

Here are some considerations:

Section 2.2. Data Collection:

Should be put flowchart showing the progress of article selection.

-> Added a flow diagram

Section 3.2. Authors, Author Keywords, Affiliations:

Figures 2 and 3 are incorrectly titled.

-> fixed in our manuscript

Section 3.3. Journals:

Table 6 has the wrong title.

-> fixed in our manuscript

I consider that the paper can be published after these changes.

-> We again appreciate your time and help.

Respectfully,

Round 2

Reviewer 2 Report

The article suggests a current and attractive topic for academia. The effort made is evident, but the manuscript is flawed in its methodology and analysis. The latter does not demonstrate a clear contribution to the academic community. The manuscript is a statistical report and not a bibliometric analysis.

Most of the analyses are purely descriptive, making the paper of no academic value. Presenting figures or tables and describing them is not research. Instead, analyses should guide the reader to more detailed information about the manuscripts, authors, countries, and topics. Authors should provide an analysis of the content of each figure and indicate the significance of the data presented.

The authors argue, "Please note that bibliometric analysis studies are quantitative studies, not qualitative studies. Therefore, it is impossible to locate every single exposed study. If it were to do so, it has to be either a systematic review or a scoping review study. With that said, we were not able to analyze further to locate studies".

In this regard, the search for information in Scopus allows downloading the information obtained in CSV format that contains various information from the authors, titles of the documents, affiliations of the authors, and DOI, among others. Therefore, there is information related to each document. Additionally, this file can be worked on Excel to generate analyses that allow more detailed information about the study presented. Scopus presents additional information about the author's profile, institutions and others that can enrich the study. An exhaustive review is not requested because it is impossible, but it is possible to know the central relationships between the units of analysis (publications, authors, countries and journals).

It is recommended that the authors review Donthu et al. (2021). https://doi.org/10.1016/j.jbusres.2021.04.070 to better understand bibliometric analysis and the large number of analyses that can be performed.

Additionally, the "co-occurrence" analysis of keywords (figure 3) is poor. The exposed label is erroneous. It should be explained what the clusters and each node mean. Usually, the cluster determines the themes of the field of study and the nodes the topics. The authors should elaborate on the studies that are part of the cluster. This is normal in the vast majority of bibliometric studies.

Figure 4 is similar to table 3, reporting no usefulness.

Author Response

Dear Reviewer:

Thank you for reviewing our manuscript.

Additionally, the "co-occurrence" analysis of keywords (figure 3) is poor. The exposed label is erroneous. It should be explained what the clusters and each node mean. Usually, the cluster determines the themes of the field of study and the nodes the topics. The authors should elaborate on the studies that are part of the cluster. This is normal in the vast majority of bibliometric studies.

-> Please review the lines 169-174, the exposed label has been revised.

Figure 4 is similar to table 3, reporting no usefulness.

-> Please review the lines 180 -184.

We appreciate your time and help,

Respectfully